# Thioredoxins: Emerging Players in the Regulation of Protein S-Nitrosation in Plants

**DOI:** 10.3390/plants9111426

**Published:** 2020-10-24

**Authors:** Tereza Jedelská, Lenka Luhová, Marek Petřivalský

**Affiliations:** Department of Biochemistry, Faculty of Science, Palacký University, Šlechtitelů 27, 78371 Olomouc, Czech Republic; tereza.jedelska@upol.cz (T.J.); lenka.luhova@upol.cz (L.L.)

**Keywords:** denitrosation, nitric oxide, S-nitrosation, plant redox signaling, reactive nitrogen species, thioredoxin, thioredoxin reductase

## Abstract

S-nitrosation has been recognized as an important mechanism of ubiquitous posttranslational modification of proteins on the basis of the attachment of the nitroso group to cysteine thiols. Reversible S-nitrosation, similarly to other redox-based modifications of protein thiols, has a profound effect on protein structure and activity and is considered as a convergence of signaling pathways of reactive nitrogen and oxygen species. This review summarizes the current knowledge on the emerging role of the thioredoxin-thioredoxin reductase (TRXR-TRX) system in protein denitrosation. Important advances have been recently achieved on plant thioredoxins (TRXs) and their properties, regulation, and functions in the control of protein S-nitrosation in plant root development, translation of photosynthetic light harvesting proteins, and immune responses. Future studies of plants with down- and upregulated TRXs together with the application of genomics and proteomics approaches will contribute to obtain new insights into plant S-nitrosothiol metabolism and its regulation.

## 1. Introduction

Protein posttranslational modifications (PTMs) evolved as a mechanism to expand the coding capacity of eukaryotic genomes to generate an enormous diversity of corresponding proteomes. PTMs comprise highly divergent forms of protein modifications, including covalent additions of small or complex chemical groups or polypeptides to protein side chains, chemical modifications of amino acids, or proteolytic cleavage. Currently, over 300 different types of PTMs have been identified, which are known to regulate many aspects of cellular functions, such as protein stability, activity, and localization, as well as signal transduction and metabolic pathways [1,2]. In plants, an array of well-studied PTMs regulates the rates of key metabolic processes of photosynthesis, respiration, photorespiration, and glycolysis [3]. Besides their role in plant development and growth, multiple PTMs regulate proteins involved in plant responses to stress conditions [4,5]. Important advances in the research of PTMs in plant physiology and biochemistry have been recently achieved by the implementation of proteomics tools; however, the understanding of how PTM mechanisms are integrated within plant cells and tissue signaling and regulation is still limited [6]. Among a chemically diverse array of PTMs, protein regulations via the redox state of structural or catalytic protein cysteines have emerged as a major regulatory mechanism in signal transduction [7,8]. Oxidative and/or nitrosative modifications of cysteine thiols, which comprise reversible intra- and intermolecular disulfide bonds (S–S) and sulfenylation (-SOH), irreversible sulfinylation (-SO_2_H), and S-sulfonylation (-SO_3_H), lead to changed biological functions and have a critical role in maintaining cellular functions under diverse growth conditions [9,10].

Nitric oxide (NO), together with hydrogen sulfide and carbon monoxide, belongs to gaseous messengers involved in multiple physiological and pathological processes across all kingdoms. NO and NO-derived reactive nitrogen species (RNS) have been recognized to play crucial roles in plant signaling and regulation of plant growth; development; and responses to abiotic and biotic stress stimuli, such as drought, salinity, high or low temperatures, and pathogen infection [11,12,13,14,15]. Signaling functions of NO in plants are mediated by three key NO-dependent PTMs, namely, metal nitrosylation in metalloproteins, S-nitrosation of cysteine thiols, and tyrosine nitration [16,17,18,19,20]. Besides this, NO signaling pathways are involved also in the control of other PTM mechanisms in plants, including acetylation, persulfidation, phosphorylation, and SUMOylation [21]. In this review, we focus on the regulatory mechanisms of protein S-nitrosation and denitrosation within NO signaling in plants. We highlight the key role of the cytosolic thioredoxin-thioredoxin reductase system (TRXR-TRX), acting actively in the denitrosation of specific protein targets. We also review how other systems of redox control, including peroxiredoxins, sulfiredoxins, and S-nitrosoglutathione reductase (GSNOR) complement the specific functions of cytosolic thioredoxins (TRXs) in protein denitrosation in plants. These highly conserved enzymes regulate multiple NO-dependent signaling pathways, protect plants from nitrosative stress conditions, and play crucial roles in plant immunity to microbial pathogens.

## 2. Origin and Fate of Thiol S-Nitrosation in Plant Cells

The metabolism of S-nitrosothiols in aerobic cells, similarly to other reactive nitrogen species (RNS), is tightly interconnected with production and degradation pathways of NO and its main reaction partners, reactive oxygen species (ROS) and cellular thiols; moreover, S-nitrosothiol formation and degradation are strongly influenced by the presence of transition metals such as copper or iron [22,23]. NO can be produced by oxidative or reductive pathways in enzyme-catalyzed or non-enzymatic reactions. In animals, constitutive and inducible isoforms of nitric oxide synthase (NOS; 1.14.13.39) represent the major enzyme source of NO. Molecular mechanisms of NO generation in plant cells and tissues were recently reviewed in detail elsewhere [12,24]. Although NOS-like enzymatic activities have been described in plants [25,26], no gene or protein with significant homology to animal NOS has been found in plants thus far, unlike unique NOS enzymes identified in some algal species [27,28]. It is assumed that the observed NOS-like enzyme activity in plants might be carried by an unknown multiprotein complex that can generate NO from L-arginine with similar cofactor requirements to mammalian NOS, except for L-tetrahydrobiopterin absent in plants [26]. Enzymatic reduction of nitrite by cytosolic nitrate reductase (NR; EC 1.6.6.1) is thus the most well-characterized NO source in plants. Other well-described pathways of NO production in plants include nitrite reduction in the electron transport chain of mitochondria and chloroplasts or non-enzymatic nitrite reduction in the acidic apoplast compartment [12,20].

Reversible modifications of cysteine thiols by RNS have emerged as important redox-based PTMs and as an integral part of NO signaling, implicated in regulation of a variety of cellular processes, including regulation of ion channels, metabolic and signaling enzymes, cytoskeleton organization, mitochondrial respiration, and gene expression [29,30,31]. S-nitrosothiols are produced by the addition of the nitrosyl (NO^.^) or nitroso (NO^+^) moiety to the sulfur atoms of cysteines in low-molecular weight thiols and proteins and are considered to be a relatively stable reservoir and a transport form of NO in vivo [32,33]. Mechanisms of specificity in signaling functions of protein S-nitrosation and denitrosation within plant redox PTMs were recently reviewed in detail elsewhere [19,34,35]. Briefly, S-nitrosothiols can originate from S-nitrosylation reaction of NO radical with thiyl radicals formed under specific circumstances in protein cysteines; however, on a quantitative scale, the major portion of S-nitrosothiols is supposed to result either from S-nitrosation, i.e., a reaction of nitrosonium cation (NO^+^) or dinitrogen dioxide (N_2_O_3_) with cysteine thiolates, or trans-S-nitrosation mediated by a NO^+^ group transfer between proteins or proteins and S-nitrosoglutathione (GSNO) [34]. Due to high intracellular levels of the tripeptide glutathione (GSH), S-nitrosoglutathione (GSNO) has been found to be the most abundant low-molecular weight S-nitrosothiol. GSNO has been suggested to serve as an intracellular reservoir of NO bioactivity as well as a transport form of NO, even though NO and GSNO were reported to react with different target proteins in the plant cell [36]. Besides a reductant or a transition metal-catalyzed decomposition to release NO, the GSNO reactivity towards proteins includes trans-S-nitrosation and S-glutathionylation, other important PTMs that can affect protein structure and biological activity [37,38]. GSNO reacts with cysteine residues through trans-nitrosylation and subsequent thiol-disulfide exchange. S-glutathionylation thus reversibly adds a tripeptide and a net negative charge that can result in structural and functional changes of the target protein.

Intracellular levels and localization of GSNO therefore strongly influence the level of protein S-nitrosation; however, the current knowledge on the modulation of subcellular distribution of GSNO in plants under physiological and stress conditions is still very limited [19,39]. Proteomics studies of redox signaling in plants have identified multiple redox PTMs of plant proteins, such as carbonylation and cysteine oxidation, glutathionylation, persulfidation, and nitrosation. Redox PTMs have been thus far mostly located in the cytosol, chloroplasts, mitochondria, peroxisomes, and nucleus, but recent studies have confirmed the plausibility of redox PTMs including S-nitrosation occurring also in extracellular proteins, including cell wall-associated proteins [10].

Although the concept of protein S-nitrosation as a biologically relevant PTM has gained wide acceptance, many crucial questions still remain to be answered as to how and to what extent protein S-nitrosation and denitrosation proceed in vivo [40]. For a long time, missing specific pathways for protein S-nitrosation questioned its plausible function as a directed signaling process. In the absence of specific enzymes, during de novo S-nitrosothiol formation, S-nitrosation targets would be determined primarily by mechanisms based on the local chemical environment, including thiol pKa values and incidence of hydrophobic amino acids, localization of NO source, and steric accessibility of S-nitrosation site [23,41]. It has been suggested that the nontargeted S-nitrosation, namely, of abundant GSH, might serve to generate a substrate pool of S-nitrosothiols, whereas the key control and specificity would be provided through subsequent transnitrosation or denitrosation reactions [42,43]. Specific S-nitrosation of a particular set of protein thiols can be provided by protein–protein transnitrosation reactions, described in mammalian systems for a rather limited number of protein pairs, e.g., TRX-caspase 3, hemoglobin–anion exchanger, or glyceraldehyde phosphate dehydrogenase-histone deacetylase 2 [44]. However, a recent report provided long-sought evidence of the “nitrosylase” (or S-nitrosothiol synthase) enzyme, at least in the bacteria *Escherichia coli*, where a large interactome of proteins that generate NO, synthesize S-nitrosothiols, and propagate signaling by trans-nitrosation, was found to regulate cell metabolism and motility [45]. Surprisingly, protein S-nitrosation as a regulatory PTM has been recently questioned, suggesting it could be only a minor and transitional cysteine modification, which would generate more stable disulfide bonds after rapid reactions with abundant cellular thiols [46]. Effectively, the conclusion that disulfides, not S-nitrosothiols, might be dominant end effectors of nitrosative signaling requires further experimental examinations.

On the other hand, the specificity and selectivity of denitrosation mechanisms related to the control of protein functions by these reversible PTMs have been challenged. From a chemical point of view, both low-molecular weight and protein S-nitrosothiols are in general relatively unstable compounds that can decompose in the biological environment by fast reactions accelerated by the presence of transition metals such as copper and iron, by reductants such as ascorbic acid, or by UV light [23]. GSH occurring in high levels in multiple cellular compartments can effectively mediate protein denitrosation through transnitrosation reactions [47,48]. Therefore, due to the reversibility of cysteine modifications, intracellular levels of S-nitrosated proteins can be directly regulated, at least in part, by intracellular GSH and GSNO levels and their time- and site-specific modulations. However, studies in mammalian models showed that a subset of proteins was resistant to GSH-mediated denitrosation, where stable S-nitrosation was suggested to result from protein conformational changes, which might shield the nitrosothiol group from denitrosation by cytosolic reductants [49]. Tubulins, glutathione-S-transferase pi, creatine and pyruvate kinases, hemoglobin, and peroxiredoxin 6 were found among proteins resistant to GSH-mediated denitrosation. Furthermore, specific enzyme systems involved in direct or indirect mechanisms of S-nitrosation control have been identified, which will be discussed here in more detail.

## 3. S-Nitrosoglutathione Reductase Indirectly Regulates Protein S-Nitrosation Status in Plants

S-nitrosoglutathione reductase (GSNOR) is an evolutionary conserved cytosolic enzyme that catalyzes an NADH-dependent reduction of GSNO, leading to the formation of glutathione disulfide (GSSG) and hydroxylamine (Figure 1) [50]. By regulating GSNO levels through its irreversible degradation, GSNOR plays a critical role in the overall metabolism of RNS, in the homeostasis of intracellular levels of NO, and control of trans-nitrosation equilibrium between low-molecular weight and protein S-nitrosothiols [51,52]. According to the current enzyme classification, GSNOR belongs to class III of Zn-dependent medium-chain alcohol dehydrogenases (ADH3; EC 1.1.1.1); however, since GSNO has been uncovered as the most effective substrate of this enzyme both in vitro and in vivo, the enzyme designation as GSNOR has widely extended within the scientific literature.

Detailed analysis of plant GSNOR structures using recombinant proteins from Arabidopsis (AtGSNOR) or tomato (*Solanum lycopersicum*; SlGSNOR) confirmed the enzyme structural similarity to its animal homologues. Plant GSNORs are homodimeric enzymes consisting of two 40-kDa subunits containing one big catalytic and one small coenzyme-binding domain with an active site localized in a cleft between the two domains [53]. Comparative genomic analysis revealed eukaryotic GSNORs as highly conserved and unusually cysteine-rich proteins [54]. Plant GSNOR activity can be regulated through S-nitrosation and oxidation of conserved cysteines [55,56,57]. The compound N6022, a pyrrole-based GSNOR inhibitor developed in human clinical studies, was confirmed to be an efficient non-competitive inhibitor of the plant enzyme at nanomolar concentrations and thus can be exploited to target plant GSNOR functions by a pharmacological approach [53,58].

GSNOR has been studied and characterized in multiple plant species, including important model plants and crops [52]. GSNOR is involved in numerous developmental processes and metabolic programs in plants via direct and indirect regulatory pathways of RNS homeostasis. In addition to its direct role in GSNO catabolism, it may control the cellular redox state by affecting intracellular levels of NADH and GSH. Currently available experimental evidence delineates the importance of GSNOR in plant responses to diverse abiotic and biotic stress conditions [52,59,60]. Stress-induced modulations of GSNOR belong to important components of the ROS and RNS signaling cross-talk, mediated by nitrosative modifications of several key enzymes of their metabolism.

Collectively, GSNOR has been recognized as a crucial element in the regulation of the protein S-nitrosation status mediated by GSNO levels as a result of time- and site-specific interplay with other enzymatic and non-enzymatic reactions controlling local levels of NO, RNS, and glutathione. GSNOR does not act directly on protein S-nitrosothiols and current evidence suggests that GSNOR functions rather as a global and poorly specific regulator of protein S-nitrosothiols through locally controlled GSH/GSNO concentration ratios. It is evident that potentially highly specific and efficient direct protein denitrosation performed by the TRXR-TRX system occupies a prominent role in multiple processes related to the regulation of protein structure and activity by post-translational modifications of protein cysteine residues (Figure 1).

## 4. The Key Role of the Mammalian Thioredoxin System in Protein Denitrosation

In mammalian cells, the cytosolic and mitochondrial TRX-TRXR system together with the glutathione/glutaredoxin (GSH-GRX) system exert decisive control over the cellular redox environment. Increased production of ROS oxidizing protein thiols and its balance by TRX- and GRX-dependent reactions have a wide range of functions in cellular physiology and pathological conditions [61]. TRXs are small proteins of ca 12 kDa containing two cysteines in their active site, which act on target proteins by their thiol-disulfide oxidoreductase activity connected with the concomitant formation of inter- and intramolecular disulfide bonds within TRX or between TRX and its target, respectively. Mammalian cells possess two TRX isoforms, cytosolic TRX1 (12 kDa), translocated to the nucleus under conditions of oxidative or nitrosative stress, and mitochondria-targeted TRX2 (18 kDa). The other component of TRX-dependent system, NADPH-dependent TRXR (EC 1.8.1.9), belongs to high-molecular weight selenoenzymes in animals. A well-described role of the TRXR-TRX system is to provide electrons to peroxiredoxins (PRXs), thiol-dependent peroxidases involved in ROS and RNS removal with a fast reaction rate, as well as in redox signaling. GRXs are specifically involved in the deglutathionylation of proteins modified by the formation of a mixed disulfide bond between glutathione and cysteine [62].

A putative chemical reason for the presence of selenocysteine in enzymes that explains the biological pressure on the genome to maintain the complex mechanisms of selenocysteine insertion has been addressed in multiple studies. The requirement of selenium in the active site of mammalian TRXR can be explained by its higher nucleophilicity compared to cysteine, lower pK_a_ of selenol as a leaving group, or higher selenium electrophilicity relative to sulfur. However, experimental evidence suggests the importance of the ability of selenoenzymes to resist inactivation by irreversible oxidation, provided by the superior ability of the oxidized form, seleninic acid, to be reduced back to selenocysteine, compared to cysteine regeneration from cysteine sulfinic acid [63].

A pivotal study of Nikitovic and Holmgren [64] reported a novel catalytic activity of TRX, showing it to be capable of cleaving the S-nitrosothiol bond of GSNO in vitro. The TRXR-TRX system was later confirmed as a functional caspase-3 denitrosylase in vivo in human lymphocytes [65]. In the reaction mechanism of protein denitrosation, the cysteine residue closer to the TRX N-terminus displaces NO by a heterolytic cleavage from the target cysteine, leading to NO release and the formation of a mixed disulfide between TRX and its target protein substrate. This mixed disulfide is subsequently decomposed by a reducing action of the thiol group of the second cysteine in the TRX active site, releasing the protein substrate with reduced cysteine and the oxidized form of TRX (Figure 1) [48]. A substrate trapping technique, which explores the stability of the mixed disulfide intermediate using a TRX mutated in the second active site cysteine, has been developed to identify S-nitrosated protein targets of mammalian TRXs [66].

TRX catalyzes the denitrosation of S-nitrosocaspase 3 and S-nitrosometallothionein with concomitant generation of nitroxyl (HNO), the one-electron reduction product of NO [67]. It is known that TRXR-deficient HeLa cells with decreased TRXR activity denitrosate S-nitrosothiols less efficiently. In HepG2 cells, TRX was found to catalyze the denitrosation of all S-nitrosoproteins with a molecular mass of 23–30 kDa [68].

Interestingly, human cytosolic/nuclear TRX1 was reported as a target of S-nitrosation with separate signaling pathways under different cellular redox states—reduced TRX is nitrosated faster and selectively at Cys62, whereas oxidized TRX was found nitrosated only at Cys73 [69]. TRX1 has been shown to transnitrosate proteins through stepwise oxidative and nitrosative modifications of specific cysteines, suggesting that TRX1 regulates its target proteins via alternating modalities of reduction and nitrosation [4]. Human TRX1 in the disulfide form can be nitrosated at Cys73 and transnitrosate target proteins, including caspase-3. Thus, similarly to GSH, which can transnitrosate proteins mediated by the formation of S-nitrosoglutathione (GSNO), TRX can either denitrosate or nitrosate proteins depending on its oxidation state [70].

The selenocysteine residue in the active site of TRXR confers the enzyme highly susceptible to nitrosative inhibition. TRXR1 in HeLa cancer cells is sensitive to a nitrosation-dependent inactivation, which can be reversed by GSH [71]. These findings point to the utmost importance of selenoprotein TRXRs in RNS-mediated signaling processes and responses to nitrosative stress [72]. In summary, NO and the TRX system show a complex interplay within the redox regulation of mammalian cells, where TRXs play an active role in attenuating NO signaling and responses to nitrosative stress, whereas NO reciprocally modulates the redox activity of TRX and TRXRs. Furthermore, TRX-related protein of 14 kDa (TRP14), a highly conserved and ubiquitously expressed oxidoreductase that efficiently reduces L-cystine, was recently reported also to reduce S-nitrosated or persulfidated protein cysteines, thereby potentially modulating both NO and H_2_S signaling [73].

## 5. Thioredoxin Systems in Higher Plants

TRXs and GRXs in higher plants are involved in the control of plant metabolism, development, phytohormone pathways, and responses to abiotic and biotic constraints [74,75]. TRXs are known to regulate embryogenesis, mobilization of seed reserves, chloroplast development, and carbon metabolism. These proteins form part of large gene families, and their outstanding diversity indicates either a high level of redundancy or functional specialization. Around 50 genes encoding TRX and TRX-like proteins and around 30 genes encoding GRXs have been identified in terrestrial plants. Plant TRXs and GRXs have been proposed to act as sensors or transmitters in relaying redox signals among plant compartments by transient and reversible protein PTMs such as disulfide bond formation, glutathionylation, or nitrosation. Biochemical properties of specific TRXs and GRXs indicate a strong specificity toward their target proteins, particularly within plastidial TRXs and GRXs. Previous studies using genetic approaches involving mutants defective in the synthesis of glutathione or one of the numerous plant GRX and TRX genes demonstrated their essential roles in thiol-based mechanisms of redox homeostasis during the plant developmental cycle. TRX and GRX functions are linked to temporal or spatial variations in ROS and RNS concentrations in specific subcellular compartments [75].

The regeneration of oxidized TRXs and GRXs in plant cells is accomplished through distinct pathways (Figure 2). Whereas most GRXs utilize reduced glutathione as an electron donor, the reduction of TRXs is more complex and dependent on their cellular localization. Cytosolic, mitochondrial, and nuclear TRXs are reduced by NADPH-thioredoxin reductases (NTR), enzymes with a flavine cofactor and a double Cys motif in the catalytic center. Distinctively, chloroplastic TRXs are reduced in a light-dependent manner by a ferredoxin-dependent thioredoxin reductase (FTR), an iron-sulfur (Fe–S) protein transferring electrons from the photosystem I into a thiol-reducing cascade [62].

Accumulated evidence endorses the prominent role of TRXs and TRXR as key components in plant protection from oxidative damage [76]. TRXs supply reducing power to reductases, which are involved in the degradation of lipid hydroperoxides or reparation of oxidized proteins, including peroxiredoxins, glutathione peroxidases, and methionine sulfoxide reductases. In contrast to animals, it has been demonstrated that cytosolic (NTRA) and mitochondrial (NTRB) reductases are non-essential in plants. Arabidopsis *ntra ntrb* double mutants are viable and fertile, showing reduced growth, decreased pollen, and a wrinkled seed phenotype [77]. Interestingly, under normal growth conditions, glutathione was able to complement the absence of NTRs in the *ntra ntrb* mutant plants, including the reduction of the most abundant cytosolic TRX h3 isoform. Despite their apparent functional redundancy, cytosolic and mitochondrial NTRs might be involved in mechanisms of plant responses to stress conditions. Cytosolic NTRA was reported to confer tolerance to oxidative and drought stresses through regulation of ROS levels [78]. Methyl viologen, an inducer of ROS accumulation and oxidative damage, strongly increased *NTRA* transcripts. Moreover, NTRA-overexpressing plants showed higher survival rates and diminished ROS induction under oxidative stress compared to wild-type and *ntra* knockout lines. Besides isoforms localized to the cytosolic, mitochondrial, and plastidial compartments, several TRXs and GRXs isoforms in Arabidopsis have been assigned to the nucleus [79]. The cytosolic NTRA was found to localize to the nucleus, where it can reduce TRXh2, TRXo1, and also nucleoredoxin (NRX1), a TRX homologue with disulfide reductase activity.

## 6. TRX Role in Denitrosation of Plant Proteins

Thus far, the functions of TRXR-TRX systems in protein denitrosation in plants have been investigated in root development [80], regulation of the translation of proteins of light harvesting complexes [81], and plant immunity [82]. Seemingly, plant TRX enzymes are capable of catalyzing the denitrosation of protein S-nitrosothiols by a mechanism different to their mammalian counterparts, one that does not involve the formation of the mixed disulfide intermediate but a transnitrosation transfer of the NO group from the protein substrate to the enzyme active site, as observed for Arabidopsis TRXh5 [83]. Interestingly, a microarray analysis found six cytosolic GRXs and an atypical chloroplastic TRX upregulated in the leaves of Arabidopsis GSNOR T-DNA insertion mutants, hypothetically to reverse increased nitrosative stress caused by the absence of GSNOR activity and S-nitrosothiol accumulation [54].

In plant roots, the intracellular redox status controlled through TRX- and GSH-dependent thiol reduction is known to determine developmental processes through modulations of auxin signaling. Using Arabidopsis double TRXR (*ntra ntrb*) and GSH biosynthesis (*cad2*) mutant lines, researchers showed that the TRX and GSH pathways altered both auxin transport and metabolism [80]. Later, auxin-induced denitrosation and partial inhibition of cytosolic ascorbate peroxidase (APX1) in Arabidopsis seedlings were found mediated by the TRXR-TRX system [84]. Moreover, levels of protein S-nitrosation were increased in roots of *ntra ntrb* mutants or roots of wild-type plants treated with TRXR inhibitor auranofin, confirming the role of NADPH-dependent TRXR in protein denitrosation during auxin-mediated root development in Arabidopsis (Figure 1). TRXR activity is induced by high NO levels, suggesting the existence of a feedback mechanism to control protein S-nitrosation in plant cells [85].

A role of reversible protein S-nitrosation has been suggested in a fine-tuning of photosynthetic apparatus components encoded by the nuclear genome in the green unicellular alga *Chlamydomonas reinhardtii* through a regulatory circuit that controls cytosolic translation of the photosystem II light-harvesting proteins (LHCII) in response to light modulations [81]. Specific S-nitrosation of a putative nucleic acid-binding protein (NAB1), a cytosolic RNA-binding protein that represses translation of certain LHCII isoform mRNAs, decreased NAB1 activity in vitro and in vivo. NAB1 could not be denitrosated by GSH, whereas the cytosolic TRX h1 together with TRXR catalyzed NAB1 denitrosation in vitro. Of note, the regulation of NAB1 activity is rather atypical, as the enzyme is nitrosated under basal low-light conditions and de-nitrosated in response to stress stimuli, i.e., higher light intensity. Nevertheless, NAB1 represents a unique example of stimulus-induced denitrosation within photosynthetic light acclimation processes, and further studies in the plant cells could provide important new insights into the crosstalk between the chloroplast and cytosol.

Plant responses to pathogen challenges involve efficient recruitment of specific TRX isoforms that can regulate diverse signaling and effector components of plant immunity [35]. NPR1 (NON-EXPRESSOR OF PATHOGENESIS-RELATED GENES1), a master regulator of salicylic acid (SA)-mediated defense genes, is sequestered into the cytoplasm as an oligomer through intermolecular disulfide bonds. During SA-mediated immune responses, NPR1 monomers translocate to nuclei, where they activate pathogenesis-related gene expression. The SA-induced NPR1 oligomer-to-monomer conversion is catalyzed by TRXh5 in vivo, whereas S-nitrosation of NPR1 by GSNO facilitates its oligomerization to prevent NPR1 from translocation to the nucleus [82]. GSH participates in the crosstalk with other signaling molecules in plant responses to biotic stress through the NPR1-dependent SA pathway. Significantly enhanced transcription of TRXh and GSNOR genes was found in transgenic tobacco plants overexpressing γ-glutamylcysteine synthetase with higher GSH levels, which show increased tolerance to *Pseudomonas syringae* pv. *tabaci* [86]. In this study, GSNOR, TRXh, and other genes of the SA pathway dependent on NPR1 were found to be upregulated in tobacco BY-2 cells treated with exogenous GSH.

In the plant immune response, TRXh5 was found to be an effective protein S-nitrosothiol reductase, providing reversibility and specificity to S-nitrosation signaling [83]. Data indicate that TRXh5 and GSNOR, enzymes exhibiting similar subcellular localization, might have partially distinct groups of targets among S-nitrosated proteins, thus regulating different signaling pathways within plant immunity. TRXh5 has been suggested to discriminate between protein S-nitrosothiol substrates to provide specificity and reversibility to S-nitrosation signaling, as observed in a previous study on Arabidopsis. Here, TRXh5 was not involved in GAPDH denitrosation, whereas S-nitrosated GAPDH was specifically denitrosated by GSH-mediated trans-nitrosation [87].

Interestingly, the interplay between GSNOR and TRXs and their regulation via S-nitrosation/denitrosation have been described within the biosynthesis of phenylpropanoid-derived styrylpyrone polyphenols in the medicinal mushroom *Inonotus obliquus* [88]. S-nitrosation of the key enzymes in the phenylpropanoid biosynthesis decreases their activity, which can be restored by TRX-mediated denitrosation. Moreover, TRX acts as a trans-nitrosylase, leading to S-nitrosation of GSNOR via protein–protein interaction and thus to decreased GSNOR activity. Whether similar mechanisms might operate also in higher plants remains to be explored.

## 7. Complementary Role of Plant Peroxiredoxins and Sulfiredoxins in the TRX-Dependent Denitrosation

Peroxiredoxins (PRXs) are ubiquitously occurring thiol peroxidases with important functions in plant redox signaling networks and antioxidant defense [89]. Multiple forms of PRXs with distinct subcellular localization exhibit differential specificities toward peroxide substrates and are involved in the redox homeostasis and transduction of redox signals. In plants, gene expression, post-transcriptional and post-translational regulation, and switching or tuning of metabolic pathways belong to processes very likely targeted by PRXs [62,89].

The mammalian PRX family includes of six members divided into several groups according to the number of conserved cysteine residues and catalytic mechanism: two-cysteine PRXs (2-Cys PRXs; PRX1–4), atypical 2-Cys PRX (PRX5), and 1-Cys PRX (PRX6). In the first step of the peroxidase cycle of typical 2-Cys PRXs, a conserved cysteine reacts with H_2_O_2_, forming cysteine sulfenic intermediate. A cysteine residue located on the adjacent monomer then reacts with the sulfenic group to an interchain disulfide. The oxidized dimeric form of PRX is subsequently reduced by the TRXR-TRX system. A genome-wide study identified a minimum set of six PRXs expressed in all higher plants, including one 1-CysPrx involved in plant embryogenesis; one plastidial PRX Q; one plastidial 2-Cys PRX; and one each of cytosolic, mitochondrial, and plastidial type II PRXs. However, variation with multiple isoforms are frequently observed, such as two 2-Cys PRXs and three cytosolic PRXs II in *Arabidopsis thaliana*, but two PRX II in poplar and only one in rice [90].

Plant members of the group of two-cysteine PRXs (2-Cys PRXs) were reported to show peroxynitrite reductase activity in yeast and thus to play a role in protection against reactive nitrogen species [91]. Plant cytosolic PRX IIE and IIB, and mitochondrial PRX IIF were identified as S-nitrosation targets in a proteomic study of Arabidopsis plants treated with gaseous NO [92]. The biological relevance of S-nitrosation-dependent PRX IIE inhibition was demonstrated within the signaling events of plant defense to bacterial pathogens. Levels of protein nitration, induced by infiltration of Arabidopsis leaves with the avirulent pathogen *P. syringae* pv. *tomato*, were increased in PRX IIE knockout plants, whereas protein nitration was not detected in plants overexpressing PRX IIE [93]. In human cells, it has been reported that S-nitrosocysteine blocked the TRX-mediated regeneration of oxidized PRX, caused partially by direct modulation of TRXR activity [94]. S-nitroso donors induced the S-nitrosation of PRX1, which promoted interchain disulfide formation between two catalytic cysteines, leading to oligomer disruption and loss of peroxidase activity. In the coupled PRX-TRX system studies in vitro, S-nitrosocysteine blocked the TRX-dependent regeneration of oxidized PRX1, partly due to direct modulation of TRX reductase activity. It was suggested that in animal cells, low S-nitrosothiol levels can inhibit the complete peroxidase system mainly by inhibiting TRXR, whereas the inhibitory effect of high S-nitrosothiol levels on peroxide removal is mediated also by PRX1 nitrosation. On the basis of these findings, it is intriguing to speculate that S-nitrosation might also functionally regulate the plant TRXR-TRX-PRX system.

Sulfiredoxins (SRXs) represent an enzyme family involved in the maintenance of cellular redox balance through an ATP-dependent reduction of cysteine sulfinic acid derivatives of PRXs back to sulfenic acid, which leads to reactivation of PRX peroxidase activity capable of controlling cellular H_2_O_2_ levels [95]. In human neurons, sulfiredoxin SRX1 reverses the aberrant nitrosation of PRX occurring in some neurodegenerative disorders and can serve as a master regulator of redox reactions that protect brain cells from nitrosative and oxidative stress [96]. This study observed that SRXn1-mediated denitrosation of PRX2 involves disulfide bond formation between SRX and PRX proteins, providing a structural basis for the enzymatic reaction. The PRX cysteine denitrosation required ATP hydrolysis and was proposed to proceed via an N-phosphorylation mechanism, including the formation of unstable P-nitrosophosphine oxide and its fast hydrolysis to thiol and nitroxyl.

In plants, SRX localizes to plastids and mitochondria, where SRX can play a crucial role in the regulation and regeneration of PRXs from its over-oxidized form in high-ROS producing conditions [97]. Functional TRX/PRX/SRX system as the key element of plant redox signaling and regulation may have a decisive influence on plant yield and growth under stress conditions [98]. Members of the SRX family thus represent other promising candidates to function as important components of protein denitrosation mechanisms in plastids.

## 8. Other Mechanisms of Protein Denitrosation in Plants

Beside TRXs and GSNOR, recognized as the principal direct and indirect denitrosation systems, respectively, additional enzymes have been reported to exhibit denitrosylase activity in animals [51]. Their physiological relevance and functions both in animals and plants remain, to a large extent, to be experimentally confirmed. Protein disulfide isomerases (PDIs) are molecular chaperones that contain TRX domains and catalyze the formation, breakage, and rearrangement of protein disulfide bonds in the endoplasmic reticulum, contributing to proper folding of nascent proteins. A cell surface PDI was observed to release NO from GSNO in human platelets [99]. PDIs were suggested to decompose GSNO at low NO levels, whereas during high NO levels, PDIs would act as a NO carrier through the PDI S-nitrosation [100]. A genome-wide search identified 12 or more PDI-related members in *Arabidopsis thaliana*, classified into three groups on the basis of polypeptide length, presence of signal peptide and endoplasmic reticulum retention signal, and type of TRX domains [101]. Interestingly, mammalian PDIs are known to consist of active and inactive TRX modules, where catalytically active domains contain a di-cysteine (Cys-Gly-His-Cys) sequence motif. A recent study on recombinant Arabidopsis AtPDI1 confirmed the presence of this motif in the active site of the plant enzyme [102]. On the basis of current data, it is plausible to hypothesize that plant PDIs containing the TRX active site motif might have conserved a denitrosation activity towards GSNO or some specific protein substrates.

Similarly to a non-enzymatic decomposition of S-nitrosothiols to release NO catalysed by copper ions, a reductive decomposition of S-nitrosated proteins can be catalysed by Cu,Zn-superoxide dismutases (CuZn-SODs) through a copper-dependent mechanism yielding NO and the corresponding disulfide [103]. Because wild-type SODs are relatively inefficient denitrosylases [104], they are active in protein denitrosation probably only in sites inaccessible to more efficient denitrosylases, e.g., in mitochondria absent of GSNOR activity [105]. Under elevated ROS levels, the animal superoxide dismutase 1 (SOD1) isoform rapidly relocates into the nucleus, where it acts as a nuclear transcription factor and regulates the expression of oxidative resistance and repair genes [106]. It is attractive to speculate that, through stress-induced nucleolar translocation, Cu-ZnSOD might modulate the S-nitrosation status of nuclear proteins, e.g., histone acetylases that were shown to be targets of S-nitrosation [107]. The potential role of SOD in the decomposition of plant S-nitrosothiols, therefore, requires further experimental investigations.

## 9. Conclusions and Perspectives

Great advances have been achieved in the last decade in our understanding of how redox PTMs of protein cysteines regulate plant development and responses to stress conditions. Molecular mechanisms enabling specific and selective nitrosation and denitrosation previously identified in mammals have been identified also in plant cells, with key functions of GSNOR and the TRXR-TRX system acting on specific subsets of protein targets. Besides the substrate specificity of denitrosating enzymes, the role of intracellular localization and compartmentation of S-nitrosothiol production and decomposition surely deserves further study. Contradictory reports on putative GSNOR targeting to peroxisomes, mitochondria, and nuclei require further verification. Similarly, a mitochondria-specific TRX system with denitrosylase function identified in mammalian cells [108] has not been investigated in plants thus far. Furthermore, how NADH-dependent GSNOR activity and NADPH-dependent TRX (and similarly PRXs and GRXs) systems operate within cell compartments with the same redox potential but highly varied ratios of NADH/NAD+ and NADPH/NADP+ cofactors is not understood and requires clarification. The current fast development of genomic and proteomic tools can surely contribute to further progress in the research of plant protein S-nitrosation, particularly new methods available for studies of the plant S-nitrosoproteome, such as a recently developed experimental strategy of visualization and identification of S-nitrosated proteins as TRX substrates applicable to purified proteins or plant cell extracts [109]. Direct genome editing techniques such as CRISPR/Cas9 [110] can be used to exploit advancing knowledge on S-nitrosation as a cysteine-based redox signaling and protein regulation mechanism in order to advance our understanding of the molecular mechanisms underlying increased plant resistance to abiotic and biotic stress conditions.

## Figures and Tables

**Figure 1 plants-09-01426-f001:**
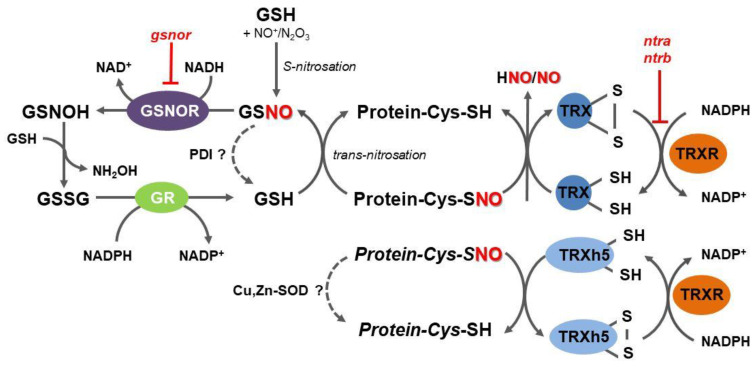
Overview of the mechanisms of protein denitrosation in plant cells. Within thioredoxin (TRX) structure, cysteine thiols present in a dithiol motif (CXXC) remove the nitroso group from S-nitrosated protein substrates, resulting in a disulfide bond formation in cytosolic TRX and free NO and related reactive nitrogen species (RNS). Reduced TRX is regenerated by NADPH-dependent thioredoxin reductase (TRXR). A specific subset of S-nitrosated proteins is denitrosated by TRXh5 in regulatory pathways of the plant immunity. A trans-nitrosation equilibrium between reduced and S-nitrosated proteins and glutathione (GSH), respectively, can be shifted by S-nitrosoglutathione reductase (GSNOR), catalyzing an irreversible NADH-dependent reduction of S-nitrosoglutathione, leading to unstable N-hydroxysulfinamide intermediate (GSNOH), which can further react with GSH to oxidized glutathione (GSSG) and hydroxylamine. GSH can be eventually regenerated by an NADPH-dependent reduction of GSSG catalyzed by glutathione reductase (GR). GSNO might be directly decomposed by protein disulfide isomerase (PDI), whereas S-nitrosoproteins might be potentially denitrosated by Cu,Zn-superoxide dismutases (Cu,Zn-SOD); *ntra ntrb*, Arabidopsis double mutant in NTRA and NTRB.

**Figure 2 plants-09-01426-f002:**
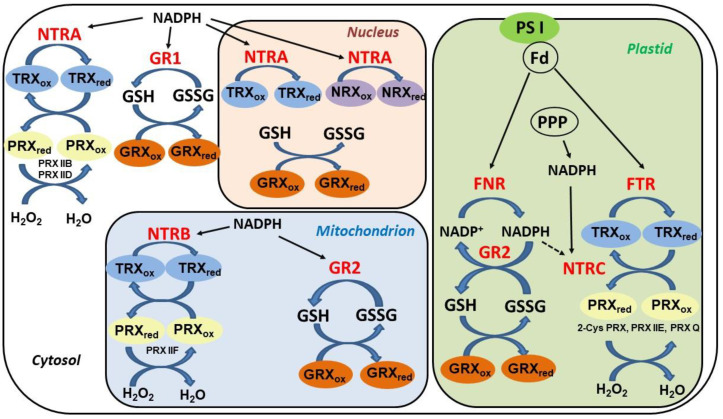
Overview of components and reduction pathways of the thiol-redox system in the main subcellular compartments of plant cells. Fd, ferredoxin; FNR, ferredoxin NADP+ reductase; FTR, ferredoxin thioredoxin reductase; GSH, reduced glutathione; GSSG, oxidized glutathione; GR, glutathione reductase (types 1 and 2); GRX, glutaredoxin; NTR: NADPH thioredoxin reductase (three types: A, B, and C); TRX, thioredoxin; PPP, pentose phosphate pathway; PRX, peroxiredoxin; PS I, photosystem I.

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
