# Peer review of "Thioredoxins: Emerging Players in the Regulation of Protein S-Nitrosation in Plants"

_plants, 2020, doi:10.3390/plants9111426_

Round 1
Reviewer 1 Report
The manuscript by Jedelska and colleagues aims to review the current knowledge regarding the participation of plant thioredoxins (TRXs) in the regulation of protein S-nitrosation, which is a critical post-translational modification altering enzyme activity and protein conformation in all living organisms. The participation of thiol reductases including TRXs in this process is relatively less documented in plants. However, the knowledge in this field has recently increased. Thus, the objectives of the review are original and relevant. Nonetheless, the present version has to be improved. Notably, the sections dedicated to TRXs and related proteins, for which the authors are less expert compared to NO biology, need substantial revision to get a better overview on plant thiol reductases, and their possible roles in protein nitrosation.
Main comments:
1- Organization and structure of the review.
The section 2. of the review is well documented and written. Then the construction of the review appears rather weak and even confuse for the reader. There is no connection between sections 2 and 3. I suggest the authors to reorganize their manuscript in the following way. The authors mention GSH and GSNO in Section 2, but describe the GSNOR enzyme, which is a relatively well-known actor in regulation of nitrosation after the section related to TRXs. Thus, section 6, which has no obvious link with the previous one, has to be moved after section 2.
Section 7 describes the roles of other protein types in protein denitrosation. While the part related to PDIs, which are TRX-related proteins is relevant, the paragraph concerning SODs (from line 321 to 362) does not fall within the scope of the review and should be suppressed.
2- Description of TRXs and related proteins
The part dedicated to the role of the TRX systems in protein S-nitrosation needs substantial improvement. The authors first describe in a meaningful way the data gained in other organisms supporting the involvement of TRXs in nitrosation and then skip to the plant TRX system, which is described in a much too short manner. The rather important literature about TRX systems in higher plants has not been sufficiently explored (see reviews by Vieira Dos Santos and Rey TIPS, 2006; Meyer et al., ARS, 2012; Rouhier et al., BBA 2015; Geigenberger et al., TIPS, 2017; Liebthal et al., ARS, 2018). Here are some examples illustrating this comment:
- TRXs and related proteins are key actors in plant responses to environmental constraints leading to modified redox homeostasis (see review by Vieira Dos Santos and Rey, TIPS, 2006). Therefore, interplays between NO signaling and the TRX system are highly likely to occur in such adverse conditions.
- GRXs, which are closely related to TRXs, are just mentioned at the beginning of section 4. The authors should give more information on these thiol reductases having notably key roles in signaling in plants as shown by many papers in the last years.
- I would advise the authors to add a figure depicting the main TRX and PRX systems in the various subcellular compartments. This will help readers who are not familiar with the rather complex world of plant TRXs and PRX proteins, and their sources of electrons (NADPH-TRX reductase in cytosol and mitochondria and ferredoxin-TRX reductase in chloroplasts). The authors do not mention the plastidial reduction system, and do not discuss what could be the interplays with NO signaling in this compartment.
- What do the authors mean by “Complementary role of PRX and SRX in the plant TRX system”? (Line 250). PRXs fully belong to the TRX system since they exchange reducing power with TRXs. PRXs have been shown recently to fulfil key roles in redox signaling by deactivating photosynthesis at night (Yoshida et al., PNAS, 2018). This section has to be greatly improved to show the relationships between TRXs, PRXs and SRX.
Lines 189-190: the plastidial FTR-TRX reduction system has also a key role in plant protection against oxidative damage (Vieira Dos Santos and Rey, TIPS, 2006).
- Line 269: Correct reference for the first report about SRX (Biteau et al., Nature, 2003).
- Line 273: Correct reference for the first report showing SRX plastidial localization and activity in plant cells: Rey et al., Plant J (2007)
Minor points:
- Lines 15-16: incomplete sentence.
- Line 57: correct to “sulfiredoxin”.
- Line 178: suppress “Authors”
- Line 197: what is the “hot” mutant, which is not mentioned before?
Author Response
Reviewer 1
The manuscript by Jedelska and colleagues aims to review the current knowledge regarding the participation of plant thioredoxins (TRXs) in the regulation of protein S-nitrosation, which is a critical post-translational modification altering enzyme activity and protein conformation in all living organisms. The participation of thiol reductases including TRXs in this process is relatively less documented in plants. However, the knowledge in this field has recently increased. Thus, the objectives of the review are original and relevant. Nonetheless, the present version has to be improved. Notably, the sections dedicated to TRXs and related proteins, for which the authors are less expert compared to NO biology, need substantial revision to get a better overview on plant thiol reductases, and their possible roles in protein nitrosation.
Main comments:
1- Organization and structure of the review.
The section 2. of the review is well documented and written. Then the construction of the review appears rather weak and even confuse for the reader. There is no connection between sections 2 and 3. I suggest the authors to reorganize their manuscript in the following way. The authors mention GSH and GSNO in Section 2, but describe the GSNOR enzyme, which is a relatively well-known actor in regulation of nitrosation after the section related to TRXs. Thus, section 6, which has no obvious link with the previous one, has to be moved after section 2.
Reply: We appreciate this reviewer comment and as suggested, we have moved this section, originally numbered as n.6, after the section n. 2, i.e. numbered as 3 in the revised version of the MS.
Section 7 describes the roles of other protein types in protein denitrosation. While the part related to PDIs, which are TRX-related proteins is relevant, the paragraph concerning SODs (from line 321 to 362) does not fall within the scope of the review and should be suppressed.
Reply: Although TRXs and TRX-related proteins are the main focus of our review, we believe that its scope include mechanisms of nitrosation and denitrosation in general, as evidenced by the sections n.2 and 3. For this reason we have considered valuable for the readers to mention very briefly that SOD has been recognized as a component of denitrosation mechanisms in animals.
2- Description of TRXs and related proteins
The part dedicated to the role of the TRX systems in protein S-nitrosation needs substantial improvement. The authors first describe in a meaningful way the data gained in other organisms supporting the involvement of TRXs in nitrosation and then skip to the plant TRX system, which is described in a much too short manner. The rather important literature about TRX systems in higher plants has not been sufficiently explored (see reviews by Vieira Dos Santos and Rey TIPS, 2006; Meyer et al., ARS, 2012; Rouhier et al., BBA 2015; Geigenberger et al., TIPS, 2017; Liebthal et al., ARS, 2018).
Reply: We have improved the entire manuscript according reviewer´s suggestions, including the part dedicate to plant TRXs. We have included citations of suggested papers by Vieira Dos Santos and Rey TIPS, 2006 and Rouhier et al., BBA 2015, whereas the other suggested reviews had been already included within the cited literature. We would like again to stress that it has not been the aim of our manuscript to cover the entire field of plant TRX in a deep detail, rather to focus to their specific role in regulation of proteins S-nitrosation.
Here are some examples illustrating this comment:
- TRXs and related proteins are key actors in plant responses to environmental constraints leading to modified redox homeostasis (see review by Vieira Dos Santos and Rey, TIPS, 2006). Therefore, interplays between NO signaling and the TRX system are highly likely to occur in such adverse conditions.
Reply: We agree that this interplay is likely to occur; however, we could not found any further reports or published evidence on interplay of NO signalling and the plant TRX systems, beside of those already commented and cited in the manuscript, which would enable to extend this part of the manuscript.
- GRXs, which are closely related to TRXs, are just mentioned at the beginning of section 4. The authors should give more information on these thiol reductases having notably key roles in signaling in plants as shown by many papers in the last years.
Reply: We have carefully evaluated this reviewer´s comment and in our opinion, GRXs are out of scope of our review. Although GRXs indisputably play key role in redox signalling, there are no reports which would suggest any role for GRXs in denitrosation of protein cysteines. For this reason, we have considered appropriate just to introduce briefly GRXs at the beginning of the mentioned section.
- I would advise the authors to add a figure depicting the main TRX and PRX systems in the various subcellular compartments. This will help readers who are not familiar with the rather complex world of plant TRXs and PRX proteins, and their sources of electrons (NADPH-TRX reductase in cytosol and mitochondria and ferredoxin-TRX reductase in chloroplasts). The authors do not mention the plastidial reduction system, and do not discuss what could be the interplays with NO signaling in this compartment.
Reply: As suggested, we have added a new Figure showing the main TRX/GRX/PRX systems in plant subcellular compartments.
- What do the authors mean by “Complementary role of PRX and SRX in the plant TRX system”? (Line 250). PRXs fully belong to the TRX system since they exchange reducing power with TRXs. PRXs have been shown recently to fulfil key roles in redox signaling by deactivating photosynthesis at night (Yoshida et al., PNAS, 2018). This section has to be greatly improved to show the relationships between TRXs, PRXs and SRX.
Reply: We have corrected the mentioned caption to “Complementary role of plant PRX and SRX in TRX-dependent denitrosation“. As suggested, we have revised this section in respect to the reaction mechanism and the role of PRX in plants. However, it has not been our aim to present an exhaustive review covering all aspects of PRXs in plants, rather to present an overview of very limited knowledge on PRX/TRX regulation by nitrosation ans their role in the catabolism of RNS. For this reason, we eoul not consider relevant to include a citation of the article by Yoshida et al. 2018, dedicated to a specific PRX/TRX role in thiol-redox regulation in chloroplast.
Lines 189-190: the plastidial FTR-TRX reduction system has also a key role in plant protection against oxidative damage (Vieira Dos Santos and Rey, TIPS, 2006).
Reply: As suggested in previous comments by Reviewer 2, we have revised the section dedicated to plant TRX to complement it with a brief overview of plastid TRX system.
- Line 269: Correct reference for the first report about SRX (Biteau et al., Nature, 2003).
Reply: We have replaced the originally used reference to Biteau et al. 2003.
- Line 273: Correct reference for the first report showing SRX plastidial localization and activity in plant cells: Rey et al., Plant J (2007)
Reply: This reference has been corrected to Rey et al. 2007
Minor points:
Lines 15-16: incomplete sentence.
Line 57: correct to “sulfiredoxin”.
Line 178: suppress “Authors”
Line 197: what is the “hot” mutant, which is not mentioned before?
Reply: We could not identify an incomplete sentence in lines 15-16. Other issues have been corrected as suggested.
Reviewer 2 Report
The m/s contains a rather brief review of the role of certain proteins in modulating plant protein s-nitrosation/denitrosation and I find the title misleading since less than 20% of the paper talks about an eventual role for thioredoxins (TRXs) in this redox PTM that reflects the rather limited amount of evidence available to show that TRXs are important players in plants. Therefore, I suggest that the title should be changed to give a better description of the overall content. Accordingly the abstract should be modified and it should also be specified that it concerns the emerging role of TRX-TRXR in protein S-nitration and recent advances in the properties functions and regulation of plant TRXs in root development, photosynthesis and immune responses related to this specific PTM. Furthermore, the last sentence of the abstract is confusing to me, as I am not sure whether it is related to perspectives or to the approaches used to obtain the data reviewed here?
Overall, I find the review too succinct and I believe more details could be given throughout (some examples are mentioned below) that would be beneficial to non-specialists. The overall feeling I have is that this is mainly a review of review papers and not original research papers and this perhaps leads to the lack of details. For a review proposed to be highlighting TRXs, I feel that plant TRX systems are not fully introduced. Finally, the paper contains a single figure and it might benefit from additional figures to give the reader a visual aid to help understand some of the information given.
Some minor points, suggestions and questions that the authors might like to take into consideration to perhaps improve their m/s:
Line 33: perhaps more types of PTM could be mentioned, I find phosphorylation and disulfide bonds rather limited. Lines 32-35: This could be better written, perhaps as 3 sentences..
Line 49: Perhaps include which abiotic and biotic stresses.
Line 55: “the TRX-TRXR system” – the use of “the” suggests only one such system, whereas in plants this is rather complex even though the limited data available point to cytosolic Trxh forms with respect to S-nitrosation (which are of course linked to NTR and not Fd-dependent TRX reductase).
Line 56: Which specific functions of TRXs in plants are complemented with other redox control systems?
Part 2 compares mammalian NO generation via NOS with what is believed to be taking place in plant systems that lack NOS except for some algal species. Perhaps more could be said about the algal NOS-like proteins.
Line 80: “a variety of cellular processes” - but which ones?
Line 100: “redox protein PTMs” – which PTMs are included here?
Line 101: what about peroxisomes?
Line 126: “On the other side” should be changed to “On the other hand”.
Line 135: Does this sentence refer to plants or mammals or both? Can the “numerous proteins” be given in the text (or some examples)?
In this part it might be nice to have a Figure showing the different reactions mentioned to produce S-nitrosothiols, perhaps show the structure of GSNO.
Part 3 is dedicated to mammalian systems and therefore perhaps this should be mentioned in its title. TRXs are introduced here but very briefly and only NADPH-TRXR is mentioned. This is far too simplistic with respect to plants where TRXs are found in several compartments, where there are several different forms often composed of isoforms and of course there is also the Fd-TRX reductase in plastids. Furthermore, I believe that plant TRXR is not a selenoenzyme. The role of TRXs is not uniquely to give electrons to thiol-dependent peroxidases involved in ROS and RNS removal. Surely, in a “plant” review the authors should describe the TRX-TRXR systems in plants and their known roles.
Line 162: “human TRXs” – how many are there? Only TRX1 is mentioned afterwards.
Part 4 describes several articles that suggest a role for the TRX-TRXR system in plant protein denitrosation although Ref 63 describes the interplay between TRX reduction of NPR1 and nitrosation via GSNO of NPR1 but nitrosation/denitrosation of NPR1 was not shown to be linked to TRX. In my opinion, the first paragraph lacks detail. Describe more the gene families perhaps mention specific protein targets and how their functions are altered to regulate a specific process instead of just mentioning 1 or 2 reviews.
Line 178: remove “Authors”
Line 197: Please mention what protein HOT5 encodes for .
Line 215: “the cytosolic TRXs system” – please be more specific.
Line 226: Do we know which TRXs are involved here? If so please include their names.
Part 5 is again, very succinct and based on reviews so details are not given. Why is it believed that these other systems have complementary roles in the plant TRX system? It is stated that certain PRXs are perhaps S-nitrosylated (this might be the case for certain plant TRXs but this information was omitted). The paragraph on sulfiredoxins does not appear to be linked to protein s-nitrosation.
Line 314: surely this sentence should refer to protein-s-denitrosation by the TRX system?
Lines 390-392: remove text?
Author Response
Reviewer 2
The m/s contains a rather brief review of the role of certain proteins in modulating plant protein s-nitrosation/denitrosation and I find the title misleading since less than 20% of the paper talks about an eventual role for thioredoxins (TRXs) in this redox PTM that reflects the rather limited amount of evidence available to show that TRXs are important players in plants. Therefore, I suggest that the title should be changed to give a better description of the overall content.
Reply: We agree with the point raised by the reviewer that the evidence for the role of TRXs in regulation of protein S-nitrosation is currently rather limited, and this is reflected in the “emerging” status mentioned in the title. Notwithstanding, this was one of the reason and aims of the review, to bring to the attention to researchers within plant field the overview of the knowledge available in animal field, presented in a wider context of nitrosation and denitrosation mechanisms beyond TRXs. Collectively, we consider the title appropriate in relation to the overall content of the manuscript.
Accordingly the abstract should be modified and it should also be specified that it concerns the emerging role of TRX-TRXR in protein S-nitration and recent advances in the properties functions and regulation of plant TRXs in root development, photosynthesis and immune responses related to this specific PTM. Furthermore, the last sentence of the abstract is confusing to me, as I am not sure whether it is related to perspectives or to the approaches used to obtain the data reviewed here?
Reply: We appreciate the reviewers comments and we have revised the abstract, including the last sentence, accordingly.
Overall, I find the review too succinct and I believe more details could be given throughout (some examples are mentioned below) that would be beneficial to non-specialists. The overall feeling I have is that this is mainly a review of review papers and not original research papers and this perhaps leads to the lack of details. For a review proposed to be highlighting TRXs, I feel that plant TRX systems are not fully introduced. Finally, the paper contains a single figure and it might benefit from additional figures to give the reader a visual aid to help understand some of the information given.
Reply: We have revised and improved the part dedicated to TRXs according suggestions of Reviewers 1 and 2, including more detailed introduction and coverage of plant TRXs.
Some minor points, suggestions and questions that the authors might like to take into consideration to perhaps improve their m/s:
Line 33: perhaps more types of PTM could be mentioned, I find phosphorylation and disulfide bonds rather limited. Lines 32-35: This could be better written, perhaps as 3 sentences.
Reply: For the clarity of the text in this Introduction section, we preferred to revise this sentence to remove references to any specific PTM.
Line 49: Perhaps include which abiotic and biotic stresses.
Reply: We have added several relevant examples as “drought, salinity, high or low temperatures and pathogen infection” in this sentence.
Line 55: “the TRX-TRXR system” – the use of “the” suggests only one such system, whereas in plants this is rather complex even though the limited data available point to cytosolic Trxh forms with respect to S-nitrosation (which are of course linked to NTR and not Fd-dependent TRX reductase).
Reply: At this specific point, we have added the term of “cytosolic” to clarify this fact.
Line 56: Which specific functions of TRXs in plants are complemented with other redox control systems?
Reply: We have reworded this sentence to clarify that this concerns functions in S-nitrosation control.
Part 2 compares mammalian NO generation via NOS with what is believed to be taking place in plant systems that lack NOS except for some algal species. Perhaps more could be said about the algal NOS-like proteins.
Reply: We have evaluated this reviewer´s suggestion, but we consider that out of the scope of our review; moreover, more detailed description on NO production in algae including structures of algal NOS proteins are available for interest readers in the cited papers.
Line 80: “a variety of cellular processes” - but which ones?
Reply: We have revised this sentence to include examples of key cellular processes known to be regulated by S-nitrosation.
Line 100: “redox protein PTMs” – which PTMs are included here?
Reply: We have included examples of redox PTMs in this sentence.
Line 101: what about peroxisomes?
Reply: We really appreciate this comment and we have added peroxisomes, which were inadvertently omitted, to the list of organelles, where redox PTM were investigated.
Line 126: “On the other side” should be changed to “On the other hand”.
Reply: We corrected this sentence as suggested.
Line 135: Does this sentence refer to plants or mammals or both? Can the “numerous proteins” be given in the text (or some examples)?
Reply: This sentence refers to result obtain on rat brain proteins, therefore we revised correspondingly the text and include examples of identified stable S-nitrosated proteins.
In this part it might be nice to have a Figure showing the different reactions mentioned to produce S-nitrosothiols, perhaps show the structure of GSNO.
Reply: We have evaluated this reviewer´s suggestions but we believe that there are many published reviews dedicated in general to S-nitrosation chemistry and biology which include nice schemes of known nitrosation and nitrosylation mechanisms of cellular proteins. We this reason we have not considered that including such a picture would provide any novelty in this respect in our review focused to enzyme-catalyzed denitrosation mechanisms.
Part 3 is dedicated to mammalian systems and therefore perhaps this should be mentioned in its title. TRXs are introduced here but very briefly and only NADPH-TRXR is mentioned. This is far too simplistic with respect to plants where TRXs are found in several compartments, where there are several different forms often composed of isoforms and of course there is also the Fd-TRX reductase in plastids. Furthermore, I believe that plant TRXR is not a selenoenzyme. The role of TRXs is not uniquely to give electrons to thiol-dependent peroxidases involved in ROS and RNS removal. Surely, in a “plant” review the authors should describe the TRX-TRXR systems in plants and their known roles.
Reply: As suggested in previous comments by Reviewer 1, we have revised the section dedicated to plant TRX to complement it with a brief overview of plastidial TRX system, and added a new Figure showing an overview of TRX/PRX/GRX systems in plant cell compartments. We have not suggested in our manuscript that plant TRXR would be a selenoenzyme, but we have added an explicit mention of the cofactor requirement of plant TRXR in the revised version.
Line 162: “human TRXs” – how many are there? Only TRX1 is mentioned afterwards.
Reply: Two TRX isoform are known in mammalian cells – we have added this information into the section 4 of the revised MS.
Part 4 describes several articles that suggest a role for the TRX-TRXR system in plant protein denitrosation although Ref 63 describes the interplay between TRX reduction of NPR1 and nitrosation via GSNO of NPR1 but nitrosation/denitrosation of NPR1 was not shown to be linked to TRX. In my opinion, the first paragraph lacks detail. Describe more the gene families perhaps mention specific protein targets and how their functions are altered to regulate a specific process instead of just mentioning 1 or 2 reviews.
Reply: We appreciate this reviewer´s comment. We have split the mentioned part into two part, numbered 5 and 6 in the revised MS, one to describe more detail the plant TRX-TRXR system, and the other to comment actual knowledge on the role of TRX in protein denitrosation. Regarding ref.63, we have included the reference to the results of the study by Tada et al. to illustrate TRXs functions in plant immunity in a wider context. We did not state that TRX was involved in NPR1 denitrosation, but rather that it catalysed oligomer-monomer transition of NPR1, which was shown by the authors of ref. 63 related also to its nitrosation status.
Line 178: remove “Authors”
Reply: The word “Authors” was removed.
Line 197: Please mention what protein HOT5 encodes for .
Reply: HOT5 protein is identical to GSNOR, as reported by Lee et al. 2008. We have corrected this sentence to make the message clear.
Line 215: “the cytosolic TRXs system” – please be more specific.
Reply: We have specified at this point that TRX h1 was tested in the cited study.
Line 226: Do we know which TRXs are involved here? If so please include their names.
Reply: We have specified at this point that TRX h5 was found to catalyse the release of NPR1 monomer.
Part 5 is again, very succinct and based on reviews so details are not given. Why is it believed that these other systems have complementary roles in the plant TRX system? It is stated that certain PRXs are perhaps S-nitrosylated (this might be the case for certain plant TRXs but this information was omitted). The paragraph on sulfiredoxins does not appear to be linked to protein s-nitrosation.
Reply: As suggested also by other reviewers, we have corrected the mentioned caption to “Complementary role of plant PRX and SRX in TRX-dependent denitrosation“. Furthemore, we have revised this section focused to relationships of TRX, PRX and SRX, including citation of the article by Yoshida et al. 2018. We have maintained a short paragraph on sulfiredoxin in relation to a published report showing evidence of human sulfiredoxin acting as PRX-2 denitrosylase.
Line 314: surely this sentence should refer to protein-s-denitrosation by the TRX system?
Reply: We agree and this error was corrected.
Lines 390-392: remove text?
Reply: This text has been removed.
Reviewer 3 Report
The manuscript reviews the mechanisms involved in the control of protein S-nitrosation in plants, with an emphasis on the emerging role of thioredoxins (although other players are also discussed). This is a research subject in the frontier of knowledge, that deserves such a literature review. The authors not only discussed the existing literature in plant science, but also refer to recent developments in the control of protein S-nitrosation in others organisms (as mammals, fungi, etc.), so that they provide important insights to be tested in plants. The authors also provide a clear and precise explanation about chemical aspects, which lacks for many plant scientists. The manuscript is clear and well written. I suggest some minor points to be revised.
1 – In general, the paragraphs are too long and include different ideas. I suggest the authors to decrease paragraph length, by splitting some of them.
2 - Lines 52-53: Please provide examples of other PTMs that are regulated by NO.
3 – Line 65: Correct to transition metals.
4 – Line 71: Correct to plants.
5 – Please take care with the superscripts (as in NO+, NADP+, etc.)
6 – Line 118: Please write the genera in full, as well as use italics for the scientific name.
7 – Please provide a brief explanation about glutathionylation and GSH-GRX system.
8 – Lines 146-147: It not clear where the dissulfide bonds are formed (on target protein or TRX?). Please rephrase it.
9 – Line 155: It is only implicit that free NO is released. Please make it explicit as in the Figure legend.
10 – Line 166: Remove Authors?
11 – Please define these abbreviations in their first appearance: NAB1, SA, ER, NPR1.
12 – In item 4, the authors make important reference to the interaction between GSNOR and TRXs. However, they explain GSNOR reaction and functions only later in item 6. Please consider to rearrange the text order.
13 – Line 245: Non-italicized scientific name.
14 – Line 284: It seems something is lacking here. low molecular-weight…
15 – The authors could refer to Figure 1 along the text, as it helps the reader to understand the explained mechanisms (or at least do not show it only in the end of the manuscript).
Author Response
Reviewer 3
The manuscript reviews the mechanisms involved in the control of protein S-nitrosation in plants, with an emphasis on the emerging role of thioredoxins (although other players are also discussed). This is a research subject in the frontier of knowledge, that deserves such a literature review. The authors not only discussed the existing literature in plant science, but also refer to recent developments in the control of protein S-nitrosation in others organisms (as mammals, fungi, etc.), so that they provide important insights to be tested in plants. The authors also provide a clear and precise explanation about chemical aspects, which lacks for many plant scientists. The manuscript is clear and well written. I suggest some minor points to be revised.
1 – In general, the paragraphs are too long and include different ideas. I suggest the authors to decrease paragraph length, by splitting some of them.
Reply: Whenever appropriate, we have adjusted the length of manuscript paragraphs.
2 - Lines 52-53: Please provide examples of other PTMs that are regulated by NO.
Reply: We have added the PTM examples of acetylation, persulfidation, phosphorylation, and SUMOylation in this sentence.
3 – Line 65: Correct to transition metals.
Reply: This word has been corrected.
4 – Line 71: Correct to plants.
Reply: This word has been corrected.
5 – Please take care with the superscripts (as in NO+, NADP+, etc.)
Reply: We have checked the entire manuscript for correct use of superscripts.
6 – Line 118: Please write the genera in full, as well as use italics for the scientific name.
Reply: We have corrected this issue as suggested.
7 – Please provide a brief explanation about glutathionylation and GSH-GRX system.
Reply: We have included a brief explanation on glutathionylation as a redox-based cystein PTM and also about basic functions of glutaredoxin system.
8 – Lines 146-147: It not clear where the dissulfide bonds are formed (on target protein or TRX?). Please rephrase it.
Reply: As suggested, we have rephrased the sentences to make this issue clear: “… formation of inter- and intramolecular disulfide bonds within TRX or between TRX and its target, respectively”
9 – Line 155: It is only implicit that free NO is released. Please make it explicit as in the Figure legend.
Reply: We have revised this sentence and included an explicit mention of NO released in denitrosation reaction.
10 – Line 166: Remove Authors?
Reply: This word has been removed.
11 – Please define these abbreviations in their first appearance: NAB1, SA, ER, NPR1.
Reply: We defined abbreviations NAB1, SA and NPR1 on their first appearance as suggested. We have replaced the abbreviation ER for “endoplasmatic reticulum”.
12 – In item 4, the authors make important reference to the interaction between GSNOR and TRXs. However, they explain GSNOR reaction and functions only later in item 6. Please consider to rearrange the text order.
Reply: The entire text has been rearranged as suggested also by the Reviewer 1. The section describing the role of GSNOR (originally numbered as n.6) has been moved before the section dedicated to TRXs.
13 – Line 245: Non-italicized scientific name.
Reply: This word was corrected to italics.
14 – Line 284: It seems something is lacking here. low molecular-weight…
Reply: This sentence was corrected to “low molecular-weight and protein S-nitrosothiols”
15 – The authors could refer to Figure 1 along the text, as it helps the reader to understand the explained mechanisms (or at least do not show it only in the end of the manuscript).
Reply: We have added further references to Figure 1 within the manuscript whenever appropriate.
Round 2
Reviewer 1 Report
The manuscript by Jedelska and colleagues has been substantially and satisfactorily improved. There are still some minor points that need to be addressed regarding manuscript organization, errors and writing style.
Comments:
- The Section 7 still describes the roles of two protein types in protein denitrosation, which have nothing in common regarding structure and biochemical function. While the part related to PDIs, which are TRX-related proteins is relevant in this part of the manuscript, the paragraph concerning SODs (from line 460 to 471), should be moved earlier before section 4. I understand that the authors wish to present these enzymes as possible actors in protein nitrosation, but the dedicated section is not at the right place in the present version.
The authors could modify line 188 to “by other enzymatic systems including SODs and the thioredoxin system”, and then describe the few data related to SODs, before the extensive description of the data concerning all TRX-related proteins. PDIs would thus appear in the last section as other TRX-related proteins.
- Figure 1 appears too early in the manuscript since it presents all the possible actors involved in protein nitrosation. It should be placed at the end near the conclusions. In the legend of this figure (line 196), add “cytosolic” before “TRX”.
- In Figure 2, NTRC cannot be placed at the same level as FTR. NTRC includes a TRX domain and thus directly reduces PRX.
- Throughout, the manuscript (lines 16, 57, 232, 401, 424…), the authors use the expression “TRX-TRXR system”. As the reducing power originates from TRXR, the scientific community in the field uses the expression “TRXR-TRX system”. Please correct.
- Lines 393-395: References are lacking for this important information. Add: Cerveau et al., Plant Science, 2016 and Liebthal et al. Antioxidant and Redox Signaling, 2018. Also, add “very likely” before “targeted”.
Minor points:
- Line 42: suppress “the” before “signal”.
- Line 60: replace “TRXRs” by “TRXs”.
- Lines 106-107: to avoid repetition, replace “on the distribution …modulations” by “on the modulation of subcellular distribution”.
- Line 216: replace “detected” by “present”, and suppress “of normal and tumor cells”. This information is not needed.
- Line 220; PRXs have key roles in redox signaling via direct interaction with proteins. Add after “rate”: “and in redox signaling”.
- Line 238: suppress “-“.
- Line 268: add “metabolism” after “plant”.
- Line 269: suppress “plants” and replace “pathogens” by “biotic and abiotic constraints”.
- Lines 302-303. Plastidial TRXs also participate in response to oxidative stress. Therefore suppress “and NADPH-dependent…also”.
- Line 303: delete “the” following “as”.
- Line 310: add “the” before “most”.
- Line 312: suppress “the stress”.
- Line 316: suppress “high ROS and decreased viability”.
- Line 319: it is TRXh2.
- Line 320: replace “presenting” by “having”.
- Line 325: suppress “apparatus”.
- Line 334: suppress “the” before “plants”.
- Line 341: add “of WT” after roots.
- Line 343: replace “which suggest” by “suggesting”.
- Line 350: suppress “isoform”.
- Line 355: suppress “its”.
- Line 376: suppress “the”.
- Line 386: replace “If” by “Whether”.
- Lines 403-404: replace “chloroplast” and “plastid” by “plastidial”.
- Lines 407-408: The sentence does not reflect the content of the cited paper. Therefore, add “s” to “PRX”; add “Plant” before “members”; move “in yeasts” after “activity” and suppress “plant before “protection”.
- Line 436: Generally, plants have only one SRX gene. Thus, correct to “SRX”.
- Line 441. SRX is localized in plastids. So please revise to “… mechanisms in plastids”.
- Line 446: plural for “PDI” and “isomerase”.
- Line 453: replace “composition” by “type”.
- Line 466: add “the” before “animal” and add “it” before “acts”.
- Line 481: correct to “require”.
- Line 492: the last sentence is totally out of the scope of the review and overselling. From NO to “increased yield”, there is a large gap! The authors should conclude on a more realistic objective such as better understanding plant responses to environmental constraints.
Author Response
Reviewer 1
The manuscript by Jedelska and colleagues has been substantially and satisfactorily improved. There are still some minor points that need to be addressed regarding manuscript organization, errors and writing style.
Comments:
The Section 7 still describes the roles of two protein types in protein denitrosation, which have nothing in common regarding structure and biochemical function. While the part related to PDIs, which are TRX-related proteins is relevant in this part of the manuscript, the paragraph concerning SODs (from line 460 to 471), should be moved earlier before section 4. I understand that the authors wish to present these enzymes as possible actors in protein nitrosation, but the dedicated section is not at the right place in the present version. The authors could modify line 188 to “by other enzymatic systems including SODs and the thioredoxin system”, and then describe the few data related to SODs, before the extensive description of the data concerning all TRX-related proteins. PDIs would thus appear in the last section as other TRX-related proteins.
Reply: We have carefully evaluated this reviewer´s suggestion. We understand the argument to move the section dedicated to SOD; however, we have considered more appropriate to present both PDIs and SOD in a separate chapter as mammalian enzymes known to participate in GSNO decomposition as well as protein denitrosation.
Figure 1 appears too early in the manuscript since it presents all the possible actors involved in protein nitrosation. It should be placed at the end near the conclusions.
Reply: We appreciate this reviewer´s suggestion; however, it is contradictory to suggestions by the Reviewer 2 to include citations of Figure 1 more frequently and even earlier in the manuscript. We consider that the Figure 1 is illustrative in presenting reaction mechanisms of specific enzyme systems involved in protein denitrosation, so it might be helpful for prospective readers to find the Figure 1 placed at its current location within the manuscript.
In the legend of this figure (line 196), add “cytosolic” before “TRX”.
Reply: We have added “cytosolic” into the legend of Figure 1 as suggested.
In Figure 2, NTRC cannot be placed at the same level as FTR. NTRC includes a TRX domain and thus directly reduces PRX.
Reply: We have corrected this issue in Figure 2 as suggested.
Throughout, the manuscript (lines 16, 57, 232, 401, 424…), the authors use the expression “TRX-TRXR system”. As the reducing power originates from TRXR, the scientific community in the field uses the expression “TRXR-TRX system”. Please correct.
Reply: We have corrected this issue within the entire manuscript as suggested.
Lines 393-395: References are lacking for this important information. Add: Cerveau et al., Plant Science, 2016 and Liebthal et al. Antioxidant and Redox Signaling, 2018. Also, add “very likely” before “targeted”.
Reply: We have added the term “very likely” and also two references at the end of the mentioned sentence.
Minor points:
Line 42: suppress “the” before “signal”.
Line 60: replace “TRXRs” by “TRXs”.
Lines 106-107: to avoid repetition, replace “on the distribution …modulations” by “on the modulation of subcellular distribution”.
Line 216: replace “detected” by “present”, and suppress “of normal and tumor cells”. This information is not needed.
Line 220; PRXs have key roles in redox signaling via direct interaction with proteins. Add after “rate”: “and in redox signaling”.
Line 238: suppress “-“.
Line 268: add “metabolism” after “plant”.
Line 269: suppress “plants” and replace “pathogens” by “biotic and abiotic constraints”.
Lines 302-303. Plastidial TRXs also participate in response to oxidative stress. Therefore suppress “and NADPH-dependent…also”.
Line 303: delete “the” following “as”.
Line 310: add “the” before “most”.
Line 312: suppress “the stress”.
Line 316: suppress “high ROS and decreased viability”.
Line 319: it is TRXh2.
Line 320: replace “presenting” by “having”.
Line 325: suppress “apparatus”.
Line 334: suppress “the” before “plants”.
Line 341: add “of WT” after roots.
Line 343: replace “which suggest” by “suggesting”.
Line 350: suppress “isoform”.
Line 355: suppress “its”.
Line 376: suppress “the”.
Line 386: replace “If” by “Whether”.
Lines 403-404: replace “chloroplast” and “plastid” by “plastidial”.
Lines 407-408: The sentence does not reflect the content of the cited paper. Therefore, add “s” to “PRX”; add “Plant” before “members”; move “in yeasts” after “activity” and suppress “plant before “protection”.
Line 436: Generally, plants have only one SRX gene. Thus, correct to “SRX”.
Line 441. SRX is localized in plastids. So please revise to “… mechanisms in plastids”.
Line 446: plural for “PDI” and “isomerase”.
Line 453: replace “composition” by “type”.
Line 466: add “the” before “animal” and add “it” before “acts”.
Line 481: correct to “require”.
Reply: We thank the reviewer for thorough corrections of the manuscript text. We have corrected all listed errors and sentences in the revised version.
Line 492: the last sentence is totally out of the scope of the review and overselling. From NO to “increased yield”, there is a large gap! The authors should conclude on a more realistic objective such as better understanding plant responses to environmental constraints.
Reply: We appreciate this suggestion and we have reworded the last sentence correspondingly.
Reviewer 2 Report
Although efforts have been made to answer my questions and to take into account my suggestions however I still find the manuscript lacks necessary detail and references to original research papers since cited references are heavily biased towards the citation of review papers. Certain parts, in my opinion, need to be rewritten and certain parts could be moved to a more appropriate place.
Even after revision, I still have a rather important number of questions and comments that the authors might like to take into consideration:
Lines 17-19: Perhaps remove « properties, functions and regulation » and write “advances have been recently made on the functions of plant TRXs in the control of …”.
Line 19: Change “photosynthesis regulation” to “translation of photosynthetic light harvesting proteins”.
Line 40: PMTs should be PTMs
Line 103: Please rewrite the sentence, as I do not understand why it begins with “In this reaction mechanism”.
Line 106-113: The sentence about GSNO can be moved to follow the sentence of line 102-103. The part about proteomics and PTMs could be moved to the Introduction.
Line 136-137: Should this sentence be at the end of the previous paragraph?
Line 19: Perhaps change “milieu” to “environment”.
Line 148-149: The added proteins names should be moved to line145 and inserted after “a subset of proteins”.
Line 156: Perhaps Fig.1 could be cited here. Indeed, it could be cited more often in the m/s.
Paragraph from Line 174 to Line 181: Is a good example of the lack of detail provided in this review. I ask myself, what “currently available data”, which “currently available experimental evidence”, what “stress-induced modulations” of which “several key enzymes”?
Lines 187-190: Rewrite this sentence to introduce the thioredoxin system as a potential specific and efficient protein denitrosation system. In my opinion, the second part of the sentence could be removed, as it appears to state a general role of the thioredoxin system and not its role in denitrosation.
Figure 1: Why include a specific part for Trxh5?
Line 196: This should be “Reduced Trx is regenerated by NADPH-dependent …”
Line 201: Include (GSNOH) after “intermediate”.
Please define the mutants shown in Fig.1 in the Figure legend.
Line 207: Please include mammals in the title
I still believe that there should be a paragraph about thioredoxins in plants: mentioning their structure, their activity, their diversity, known targets, their importance in regulating major plant functions. This would eventually focus more on cytosolic TRXs. I do not believe that the sentence (Lines 212-215) introduces Trxs enough. Perhaps more could be said about GSH-GRX too. There are many classes of GRX, are they all only involved in protein deglutathionylation?
Line 216: Are both Trx1 and Trx2 found in the nucleus?
Line 218: Is providing electrons to peroxiredoxins the only role of mammalian TRXs?
Paragraph (Lines 223-230): Why add this part about selenium to the review?
Line 236: Fig. 1 does not show this degree of detail in the Trx mechanism.
Lines 242-246: Please rewrite. First, because denitrosation of caspase3 has already been mentioned (Line 233), secondly because it is not clear if the HeLa cells are doubly mutated for Trx and TRXR (line244).
Line 247: It is written that “human Trxs were reported …” but only Trx1 is mentioned afterwards. Overall the modified paragraph (Lines 247-255) should be rewritten to make it clearer.
Line 263: What is TRP14?
Line 268: Which “several control points”?
Line 270: How do Trxs regulate the biogenesis of chloroplast structures? Which chloroplast structures are you referring to?
The added sentence (Lines 272-276) is not detailed enough. Line 275: How do TRXs and GRXs relay redox signals among plant compartments? Lines 276-280: Please give examples of specific Trxs and Grxs and their specific targets as well as the mutants which have demonstrated their essential roles.
Fig.2: Of course peroxiredoxin is not the only target of Trxs.
Line 302: Why include “also” in this sentence?
Line 317: Which GRXs and TRXs have been assigned to the nucleus?
Line 35: Please change “regulation of the photosynthesis apparatus” to “regulation of the translation of light harvesting proteins”.
Line 330: Please detail which GRXs and which atypical TRX. Line 331: Why remove the name of the mutant (hot5)?
Line 365: What is meant by “protein depletion”?
Line 378: GAPDH denitrosation just appears at the end of the paragraph. Is it central to what has been written in the preceding sentences?
Line 383: Which “key enzymes”?
Paragraph (Lines 390-395): This starts with a “generic” vague introduction, and lacks both detail and references. Line 402: I believe that there are more than 6 Prxs expressed in higher plants: see Dietz et al (2006) J Exp Bot 57, 1697 in which 10 are mentioned.
Line 407: Please clarify the sentence to mention that plant Prxs were expressed in yeast.
Line 473: This should be modified to “Perspectives” rather than “Conclusions”
Author Response
Reviewer 2
Although efforts have been made to answer my questions and to take into account my suggestions however I still find the manuscript lacks necessary detail and references to original research papers since cited references are heavily biased towards the citation of review papers. Certain parts, in my opinion, need to be rewritten and certain parts could be moved to a more appropriate place.
Even after revision, I still have a rather important number of questions and comments that the authors might like to take into consideration:
Lines 17-19: Perhaps remove « properties, functions and regulation » and write “advances have been recently made on the functions of plant TRXs in the control of …”. Line 19: Change “photosynthesis regulation” to “translation of photosynthetic light harvesting proteins”.
Reply: We have revised this sentence as suggested.
Line 40: PMTs should be PTMs
Reply: We have corrected this error.
Line 103: Please rewrite the sentence, as I do not understand why it begins with “In this reaction mechanism”.
Reply: We have corrected this issue.
Line 106-113: The sentence about GSNO can be moved to follow the sentence of line 102-103. The part about proteomics and PTMs could be moved to the Introduction.
Reply: The modifications and arrangement of this paragraph have been made previously based on suggestions of other reviewers, so we preferred to keep the actual version and position of the text in lines 106-113.
Line 136-137: Should this sentence be at the end of the previous paragraph?
Reply: The previous paragraph concerns with S-nitrosation mechanisms, whereas the mentioned sentence introduces a new paragraph focused on denitrosation mechanisms.
Line 19: Perhaps change “milieu” to “environment”.
Reply: As suggested, we have corrected this issue.
Line 148-149: The added proteins names should be moved to line145 and inserted after “a subset of proteins”.
Reply: The addition of the protein names at this place has been required by another reviewer so we prefer to maintain the actual version and position of this sentence.
Line 156: Perhaps Fig.1 could be cited here. Indeed, it could be cited more often in the m/s.
Reply: We have added a citation of Figure 1 in this and other places in the manuscript.
Paragraph from Line 174 to Line 181: Is a good example of the lack of detail provided in this review. I ask myself, what “currently available data”, which “currently available experimental evidence”, what “stress-induced modulations” of which “several key enzymes”?
Reply: From our point of view, GSNOR does not belong to the focus of the present manuscript and for this reason, we have just mentioned GSNOR briefly with reference to our previously published review article on this subject (see reference n. 52).
Lines 187-190: Rewrite this sentence to introduce the thioredoxin system as a potential specific and efficient protein denitrosation system. In my opinion, the second part of the sentence could be removed, as it appears to state a general role of the thioredoxin system and not its role in denitrosation.
Reply: In our opinion, it is precisely the second part of this sentence which states the role of the TRX system in denitrosation.
Figure 1: Why include a specific part for Trxh5?
Reply: We have included a specific mention of TRXh5 into Figure 1, as this is the plant TRX isoform which has been best characterized in respect to its denitrosation activity.
Line 196: This should be “Reduced Trx is regenerated by NADPH-dependent …”
Reply: We have corrected this issue.
Line 201: Include (GSNOH) after “intermediate”.
Reply: We have added the missing abbreviation as suggested.
Please define the mutants shown in Fig.1 in the Figure legend.
Reply: We have added the definition of the ntra ntrb mutant into the legend as requested.
Line 207: Please include mammals in the title
Reply: We have included “mammalian” in this title.
I still believe that there should be a paragraph about thioredoxins in plants: mentioning their structure, their activity, their diversity, known targets, their importance in regulating major plant functions. This would eventually focus more on cytosolic TRXs. I do not believe that the sentence (Lines 212-215) introduces Trxs enough. Perhaps more could be said about GSH-GRX too. There are many classes of GRX, are they all only involved in protein deglutathionylation?
Reply: We have carefully considered this reviewer comment, but in our opinion, the extent of the introduction of plant TRXs and GRXs is appropriate in respect to the focus of our review paper.
Line 216: Are both Trx1 and Trx2 found in the nucleus?
Reply: We have revised this sentence to reflect correctly the nuclear localization of TRX1.
Line 218: Is providing electrons to peroxiredoxins the only role of mammalian TRXs?
Reply: We have revised this sentence.
Paragraph (Lines 223-230): Why add this part about selenium to the review?
Reply: This paragraph has been added to the previous version of the manuscript base on the requirement of other reviewers.
Line 236: Fig. 1 does not show this degree of detail in the Trx mechanism.
Reply: We have moved the reference to Figure 1 from this place.
Lines 242-246: Please rewrite. First, because denitrosation of caspase3 has already been mentioned (Line 233), secondly because it is not clear if the HeLa cells are doubly mutated for Trx and TRXR (line244).
Reply: The first mention of TRX-mediated caspase denitrosation (line 233) is related to a historical perspective, whereas the latter sentences are concerned with the mechanism of nitrosocaspase denitrosation. We have corrected the sentence in lines 245-246 to describe correctly the experimental setting of the cited study.
Line 247: It is written that “human Trxs were reported …” but only Trx1 is mentioned afterwards. Overall the modified paragraph (Lines 247-255) should be rewritten to make it clearer.
Reply: We have corrected this sentence to mention specifically TRX1. We have also modified the sentence in lines 253-255 to make the text clearer.
Line 263: What is TRP14?
Reply: We have added the explanation of this abbreviation as suggested.
Line 268: Which “several control points”?
Reply: We have reworded this sentences.
Line 270: How do Trxs regulate the biogenesis of chloroplast structures? Which chloroplast structures are you referring to?
Reply: IN this sentence, we have replaced “biogenesis of chloroplast structures” by “chloroplast development”.
The added sentence (Lines 272-276) is not detailed enough. Line 275: How do TRXs and GRXs relay redox signals among plant compartments? Lines 276-280: Please give examples of specific Trxs and Grxs and their specific targets as well as the mutants which have demonstrated their essential roles.
Reply: We have carefully considered this reviewer comment, but in our opinion, the extent of the introduction of plant TRXs and GRXs is appropriate with respect to the focus of our review paper.
Fig.2: Of course peroxiredoxin is not the only target of Trxs.
Reply: We do not suggest that PRX are the only target of TRX. Rather, we intended to review the relationships of redox systems presented in the manuscript text in a single picture.
Line 302: Why include “also” in this sentence?
Reply: In this sentence, we refer to the role of the TRX system in plant stress responses, i.e. in addition to their roles in plant growth and development.
Line 317: Which GRXs and TRXs have been assigned to the nucleus?
Reply: Due to the complexity of the nuclear thiol redox system in plants, we preferred to include just a brief mention of its importance in this sentence
Line 35: Please change “regulation of the photosynthesis apparatus” to “regulation of the translation of light harvesting proteins”.
Reply: We have corrected this sentence as suggested.
Line 330: Please detail which GRXs and which atypical TRX.
Reply: We consider that it is not appropriate to include more details on individual
Line 331: Why remove the name of the mutant (hot5)?
Reply: The original name of the gsnor mutant has been removed from the previous version of the manuscript as requested by another reviewer.
Line 365: What is meant by “protein depletion”?
Reply: We have modified this sentence by replacing “to prevent protein depletion” by “to prevent NPR1 from translocation to nucleus”.
Line 378: GAPDH denitrosation just appears at the end of the paragraph. Is it central to what has been written in the preceding sentences?
Reply: GAPDH denitrosation it is an integral part of the paragraph, concerned with the substrate specificity of TRX denitrosation activity.
Line 383: Which “key enzymes”?
Reply: We have considered appropriate not to provide wider details on the components of phenylpropanoid biosynthesis in this mushroom.
Paragraph (Lines 390-395): This starts with a “generic” vague introduction, and lacks both detail and references. Line 402: I believe that there are more than 6 Prxs expressed in higher plants: see Dietz et al (2006) J Exp Bot 57, 1697 in which 10 are mentioned.
Reply: In this place, we have adopted the information on a minimum consensus set of plant PRX from the updated review paper of the same author (Dietz KJ. 2011. Peroxiredoxins in plants and cyanobacteria. Antioxid Redox Signal 15: 1129–1159).
Line 407: Please clarify the sentence to mention that plant Prxs were expressed in yeast.
Reply: We have corrected this sentence as suggested.
Line 473: This should be modified to “Perspectives” rather than “Conclusions”
Reply: We have modified the title to “Conclusions and perspectives”.
Round 3
Reviewer 2 Report
A few very minor comments to take into consideration:
Line153: ammonium is not shown in Figure 1
Line 195: (Fig.1) should be removed
Line 206: please inverse (TRXR-TRX) to (TRX-TRXR)
Line 215: please remove the virgule
Line 250: perhaps cytosolic/nuclear could be moved to line 245 (or it could be removed because the dual localization of TRX1 has already been mentioned)
Please use your designated abbreviations throughout the m/s: sometimes thioredoxin is written in full, as is TRX reductase etc (e.g. lines 260-261)
Line 264: change to "Thioredoxin systems in higher plants"
Line 279-290: the last part of the sentence should be removed as it has already been written earlier in the same paragraph
Line 287: remove "composed"
Figure 2: change mitochondria to mitochondrion
Line 308: remove "the stress"
Line 320: add an s to system
Line 351: change can to could
Line 356-357: I propose "During SA-mediated immune responses, NPR1 monomers translocate to nuclei where they activate pathogenesis-related gene expression.
Line 358: isoform could be removed
Line 366: oxidoreductase could be removed
Line 367: I suggest "Data indicated that ..."
Paragraph from line 390: please use PRX throughout and not Prx
Line 402: were several different yeasts studied? if not this should be changed to "yeast"
Line 477: change to nuclei
Line 483: change to "techniques" or "A genome editing technique like"
Although the m/s is well written, I noticed a problem with the use of "the" throughout (added sometimes when not necessary while absent sometimes when necessary)
Author Response
Reviewer 2
A few very minor comments to take into consideration:
Line153: ammonium is not shown in Figure 1
Reply: We have replaced “ammonium” for correct “hydroxylamine” in the corresponding text, in agreement with the reaction scheme shown in Figure 1.
Line 195: (Fig.1) should be removed
Line 206: please inverse (TRXR-TRX) to (TRX-TRXR)
Line 215: please remove the virgule
Reply: These errors have been corrected as suggested.
Line 250: perhaps cytosolic/nuclear could be moved to line 245 (or it could be removed because the dual localization of TRX1 has already been mentioned)
Reply: We have moved “cytosolic/nuclear to line 245 as suggested.
Please use your designated abbreviations throughout the m/s: sometimes thioredoxin is written in full, as is TRX reductase etc (e.g. lines 260-261)
Reply: We have checked the entire manuscript for the correct use of abbreviations TRX and TRXR.
Line 264: change to "Thioredoxin systems in higher plants"
Reply: The title has been corrected as suggested.
Line 279-290: the last part of the sentence should be removed as it has already been written earlier in the same paragraph
Reply: This part of the mentioned sentence has been removed.
Line 287: remove "composed"
Figure 2: change mitochondria to mitochondrion
Line 308: remove "the stress"
Line 320: add an s to system
Line 351: change can to could
Reply: These corrections have been performed as suggested
Line 356-357: I propose "During SA-mediated immune responses, NPR1 monomers translocate to nuclei where they activate pathogenesis-related gene expression.
Reply: We have modified this sentence as suggested.
Line 358: isoform could be removed
Line 366: oxidoreductase could be removed
Line 367: I suggest "Data indicated that ..."
Reply: These modifications have been done as suggested.
Paragraph from line 390: please use PRX throughout and not Prx
Reply: We have checked the entire manuscript for consistent use of “PRX” abbreviation.
Line 402: were several different yeasts studied? if not this should be changed to "yeast"
Reply: We have corrected this term to “yeast” as suggested.
Line 477: change to nuclei
Reply: This correction has been performed.
Line 483: change to "techniques" or "A genome editing technique like"
Reply: We have corrected the text to “techniques”.
Although the m/s is well written, I noticed a problem with the use of "the" throughout (added sometimes when not necessary while absent sometimes when necessary)
Reply: We have checked the entire manuscript for correct use of the definite article.